# Nanostructured Hybrids Based on Tantalum Bromide Octahedral Clusters and Graphene Oxide for Photocatalytic Hydrogen Evolution

**DOI:** 10.3390/nano12203647

**Published:** 2022-10-18

**Authors:** Jhon Sebastián Hernández, Maxim Shamshurin, Marta Puche, Maxim N. Sokolov, Marta Feliz

**Affiliations:** 1Instituto de Tecnología Química, Universitat Politècnica de València-Consejo Superior de Investigaciones Científicas (UPV-CSIC), Avd. de los Naranjos s/n, 46022 Valencia, Spain; 2Nikolaev Institute of Inorganic Chemistry SB RAS, 3 Akad. Lavrentiev Ave., 630090 Novosibirsk, Russia

**Keywords:** metal cluster, tantalum, graphene oxide, nanohybrid, photocatalysis, hydrogen generation

## Abstract

The generation of hydrogen (H_2_) using sunlight has become an essential energy alternative for decarbonization. The need for functional nanohybrid materials based on photo- and electroactive materials and accessible raw materials is high in the field of solar fuels. To reach this goal, single-step synthesis of {Ta_6_Br^i^_12_}@GO (GO = graphene oxide) nanohybrids was developed by immobilization of [{Ta_6_Br^i^_12_}Br^a^_2_(H_2_O)^a^_4_]·4H_2_O (i = inner and a = apical positions of the Ta_6_ octahedron) on GO nanosheets by taking the advantage of the easy ligand exchange of the apical cluster ligands with the oxygen functionalities of GO. The nanohybrids were characterized by spectroscopic, analytical, and morphological techniques. The hybrid formation enhances the yield of photocatalytic H_2_ from water with respect to their precursors and this is without the presence of precious metals. This enhancement is attributed to the optimal cluster loading onto the GO support and the crucial role of GO in the electron transfer from Ta_6_ clusters into GO sheets, thus suppressing the charge recombination. In view of the simplicity and versatility of the designed photocatalytic system, octahedral tantalum clusters are promising candidates to develop new and environmentally friendly photocatalysts for H_2_ evolution.

## 1. Introduction

The generation of molecular hydrogen using sunlight is one of the most attractive energy production pathways for the future. Within the framework of sustainable chemistry, there is a special interest in the use of photo- and electroactive materials formed by abundant elements in the Earth’s crust to obtain H_2_ from sunlight and water, as inexhaustible sources of energy. In particular, the relative abundance of tantalum (1–2 ppm) has promoted the preparation of Ta-derived photocatalysts such as Ta_2_O_5_ and tantalum (oxy)nitrides for water reduction and splitting under photocatalytic conditions [1]. Among these and other tantalum-derived materials, nanostructures (such as nanowires [2,3], nanorods [4], nanosheets [5,6], nanotubes [7,8], nanoplates [9], nanoflowers [10], and nanoparticles [11,12]) have emerged as materials with improved photo- and electrocatalytic performance in water splitting without the presence of co-catalysts. Low-nuclearity tantalum clusters appear as another kind of nanosized molecules, of which photocatalytic properties remain unexplored. Concretely, the [{Ta_6_X^i^_12_}L^a^_6_]^n+/−^ (X^i^ = halogen; L^a^ = halogen, O-donor ligand) clusters, which belong to the large family of metal atom clusters defined by Cotton [13], show interesting photo and redox properties. In their cores, 6 metal atoms of the cluster are arranged in a regular octahedral geometry, interconnected by direct metal–metal bonds, and further linked to 12 inner ligands (X^i^) and 6 apical ligands (L^a^) located at the edge-bridging and terminal positions of the tantalum cluster, respectively (Figure 1).

Various applications of the octahedral halide-bridged clusters of tantalum in the fields of crystallography, medicine, optics, and catalysis have been proposed. For instance, {Ta_6_Br^i^_12_} clusters are used as a commercial tool for phase determination of biomolecules by X-ray crystallography and as a radiographic contrast agent [14,15]. Recently, tantalum halide clusters have been employed to build nanostructured hybrid materials for solar glazing applications in energy-saving buildings [16,17,18,19]. Catalytic applications of {Ta_6_X^i^_12_} (X^i^ = Cl, Br) cluster compounds and derived composites, involving hydrogen transfer and ring closure reactions under high-temperature conditions, were reported [20,21]. The chemistry of octahedral tantalum halide clusters not only encompasses their inclusion into polymeric matrices or other molecules for obtaining the respective hybrid [16,17,18,19] or supramolecular materials [22] but also involves ligand exchange reactions, mostly in the apical positions [23]. This combination of ligand reactivity and the robustness of the cluster core, which keeps the cluster integrity intact during these transformations, favors further search for a catalytic application of this cluster family.

The preparation of tantalum bromide and chloride [{Ta_6_X^i^_12_}X^a^_2_(H_2_O)^a^_4_]·4H_2_O (X = Cl, Br) aqua clusters was developed by Koknat et al. in the 1970s [24] and recently the iodide analog was reported [25]. Because of its stability and availability, the bromide cluster (X = Br) is the most studied of the three, and its preparation, properties, and reactivity toward solvents and oxidizing agents were recently thoroughly revisited [26]. Already in the 1980s, Vogler and Kunkely showed that the bromide cluster was redox active in the photochemical reduction of water to molecular hydrogen and in the presence of HCl [27]. This property makes this cluster compound a suitable photocatalyst in the H_2_ evolution from water and sunlight. The easy ligand exchange in the [{Ta_6_Br^i^_12_}(H_2_O)^a^_6_]^2+^ cluster complex, which is the dominant species in acidic solutions of [{Ta_6_Br^i^_12_}Br^a^_2_(H_2_O)^a^_4_]·4H_2_O, is not only involved in the mechanism of the H_2_O-to-H_2_ transformation, but it is also useful for coordinative cluster anchoring onto functionalized supports for the preparation of new hybrid materials with enhanced activity in photocatalysis.

Graphene-based photocatalysts have gained increasing interest as a viable alternative to increase the yield of photocatalytic H_2_ production in solar energy conversion in chemical energy [28,29,30,31]. The most common photocatalysts, such as TiO_2_, CdS, and BiVO_4_, show low photocatalytic activities due to the low surface area and high recombination rate of photo-induced charge carriers owing to their band energies. One strategy to enhance photocatalyst reactivity for visible-light water splitting is to blend it with graphene materials in order to obtain hybrid nanostructured materials. In this sense, GO is an excellent co-catalyst in photocatalytic applications due to its high surface area, excellent electronic mobility, and high adsorption ability associated with the oxygen functionalities [29]. Among photocatalytic applications of the GO-derived composites, photocatalytic degradation of pollutants and H_2_ evolution from water has become the most relevant, but applications in the photoreduction of CO_2_, atmospheric ammonia synthesis, and photocorrossion suppression applications still need further exploration [31,32,33,34]. In mimicking the mechanism of Gratzel cells, which includes dyes as light harvesters capable of absorbing visible light and injecting electrons into the conduction band of a semiconductor material (such as TiO_2_), it is highly desirable to develop hybrid cluster-GO photocatalysts without the need of precious metals. GO serves as a support of alternative metal clusters and nanoparticles with an absorption window from the UV to the visible region. Their immobilization avoids metal aggregation and suppresses the charge recombination by channeling electrons into the GO substrates. Recently, we immobilized molybdenum octahedral clusters, which are optimal photosensitizers and electron transfer catalysts, onto GO surfaces in order to obtain nanohybrids which resulted in efficient water reduction photocatalysts [35,36]. In this work, the [{Ta_6_Br^i^_12_}Br^a^_2_(H_2_O)^a^_4_]·4H_2_O cluster was systematically immobilized on GO nanosheets by taking advantage of the labile nature of the apical ligands of the cluster and the coordinating ability of the oxygen functionalities of the GO support. The {Ta_6_Br^i^_12_}@GO hybrids developed in this work were fully characterized and used as photocatalysts for the reduction of water vapor to H_2_. Their catalytic ability in the H_2_ evolution reaction (HER) was explored under light irradiation and the synergy between the cluster and the support and its recyclability was discussed.

## 2. Materials and Methods

### 2.1. Chemicals

Tin bromide (ACS Reagent 99%), hydrobromic acid (ACS Reagent 48%), methanol (ACS Reagent ≥ 99%), tantalum(V) bromide (ACS Reagent ≥ 99%) and diethyl ether (ACS Reagent ≥ 99%), benzoic acid (ACS Reagent ≥ 99.5%), and phenol (ACS Reagent ≥ 99.0) were obtained from commercial resources (Sigma Aldrich, Merck KGaA, Darmstadt, Germany), as well as KBr (≥99% purity, Reaktiv, JSC) and gallium granulate (≥99% purity). Tetrahydrofuran (CHROMASOLV™ Plus, inhibitor-free, for HPLC, ≥99.9%, Honeywell Riedel-de Haën) was dried and deoxygenated by passing the solvents through CuO and alumina commercial columns under nitrogen atmosphere. The ultrapure water was obtained from the Milli-Q^®^ EQ 7000 Type 1 water purification system. For the photocatalytic reactions, water and methanol were deoxygenated by bubbling argon for at least half an hour. GO (solid isolated after 4000 rpm centrifugation) was prepared from natural graphite by following an optimized procedure of the improved Hummer’s synthetic method [37,38].

### 2.2. Instrumentation

Combustion chemical analysis of the samples was carried out using a Fisons EA 1108-CHNS-O analyzer (ThermoFisher Scientific, Waltham, MA, USA). Inductively coupled plasma atomic emission spectrometry (ICP-AES) analyses for the determination of atomic tantalum of the solid materials were performed after aqua regia digestion at 180 °C for 24 h in reflux, and the resulting solutions were measured in a Varian 715 spectrometer (Palo Alto, CA, USA). Fourier transform infrared spectroscopy (FTIR) spectra were measured on KBr pellets with a Nicolet 8700 Thermo spectrometer (ThermoFisher Scientific, Waltham, MA, USA). Samples were grounded with dry KBr in agate mortar and pressed by vacuum. Raman spectra were acquired from solid materials on an aluminum sample holder and under atmosphere, using a Renishaw “Reflex” spectrometer (Wotton-under-Edge, U.K.) equipped with an Olympus optical microscope. The excitation wavelength was 514 and 785 nm generated by an Ar^+^ ion laser. The laser power on the sample was 30 Mw, and a total of 10 to 40 acquisitions were taken. Spectra registered under N_2_ atmosphere were performed in a sealed reactor equipped with a quartz window. The UV-vis-NIR diffuse reflectance (DRS) were collected in the range of 200 to 2000 nm with a Varian Cary 5000 spectrophotometer. UV-vis spectrophotometric analysis for samples in solution was performed using a Varian Cary 50 UV-vis Agilent analyzer equipped with a Xe lamp as the light source and a Czerny–Turner model dual beam monochromator with 10 × 10 mm quartz cuvettes. The powder X-ray diffraction (PXRD) data of the K_4_[{Ta_6_Br^i^_12_}Br^a^_6_], [{Ta_6_Br^i^_12_}Br^a^_2_(H_2_O)^a^_4_]·4H_2_O and {Ta_6_Br^i^_12_}@GO materials were obtained with a PANalytical Cubix-Pro diffractometer equipped with a PANalytical X’Celerator detector. This equipment employed monochromatic CuKα X-ray radiation (ʎ1 = 1.5406 Å, ʎ2 = 1.5444 Å, I2/I = 0.5) and a tube voltage and intensity of 45 kV and 40 mA, respectively. It uses a variable slit with an irradiated sample area of 5 mm and the goniometer arm length is 200 mm. The diffractogram of the powder samples was obtained at room temperature in 2θ range of 2–90°. The morphology and composition of the materials were characterized by scanning electron microscope (SEM) using a ZEISS model ULTRA55 FESEM coupled to an Oxford Instruments energy dispersive X-ray (EDS) detector. Irradiation experiments were performed with a spotlight Lightnincure LC8 model, 800–200 nm, 150 W, equipped with a fiber optic light guide with a spot size of 1.0 cm diameter. Molecular hydrogen production was monitored by gas chromatography (GC) on the Agilent 490 Micro GC System, equipped with a column coated with a zeolite molecular sieve (CP-Molsieve 5Å, Agilent J&W) and a conductivity detector (TCD). Ar was taken as the carrier gas and the flow rate was set to 5 mL min^−1^. The inlet and detector temperatures in the GC run were 110 °C and 220 °C, respectively, and the isothermal oven temperature profile was set at 62 °C with an initial column pressure of 15 psi.

### 2.3. Synthesis and Characterization of Materials

The preparation of K_4_[{Ta_6_Br^i^_12_}Br^a^_6_] was performed as described by Messerle et al. with minor modifications [39]. A mixture of TaBr_5_ (6.00 g, 10.32 mmol), KBr (1.912 g, 15.92 mmol), and Ga granulate (0.64 g, 9.12 mmol) was sealed in an evacuated quartz ampule and placed into a 270 °C preheated furnace for 45 min. During this step, the ampule was taken out every 15 min. and shaken vigorously (hand protection is necessary to prevent thermal burns)—three times in total—in order to evenly spread the molten gallium. Then, the ampule was heated to 300 °C and held at this temperature for 12 h. After this procedure, the ampule was shaken to homogenize the reactants and returned into the furnace chamber. The temperature was raised to 400 °C and held for 24 h. The ampoule was removed, allowed to cool to ambient temperature, and the product was extracted to obtain [{Ta_6_Br^i^_12_}Br^a^_2_(H_2_O)^a^_4_]·4H_2_O (next step).

The [{Ta_6_Br^i^_12_}Br^a^_2_(H_2_O)^a^_4_]·4H_2_O cluster compound was synthesized as follows: 500 mg of K_4_[{Ta_6_Br^i^_12_}Br^a^_6_] was added to 3 mL of Milli-Q water and the solid material was ground in a mortar for 15 min until a homogeneous dark green mixture was obtained. The solid was extracted by filtration through a medium porosity fritted glass funnel by adding 40 mL of water in order to remove the residual GaO(OH) from the K_4_[{Ta_6_Br^i^_12_}Br^a^_6_] precursor and other impurities in the material. In order to prevent cluster oxidation under standard environmental conditions, the dark green filtrate was treated with a small portion of SnBr_2_ (*ca.* 5 mg) followed by HBr (50 mL). The solution was kept at 4 °C, and after 20 h a dark green solid precipitated. The translucent supernatant was carefully removed by decantation, the solid was washed with diethyl ether and dried by rotatory evaporation at 60 °C for 3 h to give 355 mg of a green microcrystalline material identified as [{Ta_6_Br^i^_12_}Br^a^_2_(H_2_O)^a^_4_]·4H_2_O (Yield: 71 %). This material was stored in a desiccator and characterized by UV-vis-NIR DRS, FTIR, Raman, DRX, and SEM–EDS techniques. The amount of solvation molecules was assigned according to elemental analysis of this cluster material (% *w*/*w*, found (H, 0.665); calc. H: 0.681), and to the Raman and PXRD characterizations.

The {Ta_6_Br^i^_12_}@GO-20 materials were prepared under argon using Schlenk techniques by addition of [{Ta_6_Br^i^_12_}Br^a^_2_(H_2_O)^a^_4_]·4H_2_O (10 mg) to a GO suspension (40 mg in 100 mL of THF), which was previously sonicated during 1 h in a round bottom flask. The mixture was magnetically stirred at 40 °C for 2 h or 16 h. Next, for each reaction, the solid was separated by filtration under vacuum and it was washed with THF under an inert atmosphere. The resulting products were dried under vacuum to provide black solids labeled as {Ta_6_Br^i^_12_}@GO-20S (42 mg) and {Ta_6_Br^i^_12_}@GO-20L (44 mg), “20” referring to the percentage in weight (*w*/*w*) of cluster with respect to GO, and “S” and “L”—to the shortest (2 h) and longest reaction times (16 h), respectively. This synthetic procedure was extended to {Ta_6_Br^i^_12_}@GO-5L (5% *w*/*w*) and {Ta_6_Br^i^_12_}@GO-40L (40% *w*/*w*), and the amount of the solid obtained was 41 mg and 38 mg, respectively. These materials were stored under N_2_ atmosphere in a MBraun dry box, and characterized by UV-vis-NIR DRS, FTIR, Raman, DRX, and SEM–EDS techniques. The amount of Ta was determined by ICP analyses. Elemental analyses provided the C and H content (wt %) of each sample, namely: {Ta_6_Br^i^_12_}@GO-5L (C 57.03, H 1.79), -20S (C 54.00, H 1.66), -20L (C 51.70, H 2.70), −40 L (C 36.71, H 1.33), and GO (C 48.00, H 2.22).

### 2.4. Spectroscopic Analysis of the Reaction between [{Ta_6_Br^i^_12_}Br^a^_2_(H_2_O)^a^_4_]·4H_2_O with Phenol and Benzoic Acid

Three solutions were prepared for UV-vis identification in THF: (i) the cluster/phenol solution, which was prepared by dissolving the [{Ta_6_Br^i^_12_}Br^a^_2_(H_2_O)^a^_4_]·4H_2_O (3 mg, 1.3 µmol) and phenol (0.3 g, 3.2 mmol) in THF (4 mL) in a round bottom flask; (ii) the cluster/benzoic acid solution, which was prepared following the same methodology, but using 3 mg (1.32 µmol) of cluster precursor, 3 mg (24.5 µmol) of benzoic acid, and THF (4 mL); and (iii) the control solution of the cluster dissolved in THF. All solutions were kept at 40 °C with constant stirring for 24 h. All the manipulations were performed under argon and using Schlenk techniques. An aliquot of the resulting mixtures was diluted and placed in a spectroscopic cuvette at the end of the reaction.

### 2.5. Photocatalytic H_2_ Evolution Procedure

The photocatalytic reactions were carried out in the presence of aqueous mixtures in the vapor phase and under deaerated conditions. The photoreactor was a double cylindrical quartz reactor (110 mL of total volume) in which the two vessels (1 and 2) were connected with a quartz bridge (2 cm length). Appendix A illustrates the experimental layout for vapor water photoreduction. Water (30 mL MilliQ water), HBr 0.7 M, and the sacrificial electron donor (methanol, 19.85% *v*/*v*) were loaded into the reactor vessel (1) and purged with Ar (30 min). The photocatalysts (5 mg of {Ta_6_Br^i^_12_}@GO or [{Ta_6_Br^i^_12_}Br^a^_2_(H_2_O)^a^_4_]·4H_2_O materials) were dispersed in methanol (0.2 mL), and carefully deposited in the reactor vessel (2) by drop casting with simultaneous slow evaporation under a current of Ar, until a thin film was obtained. The reactor was sealed and pressurized with argon up to 0.3–0.5 bar and connected to an electrical heating ribbon that allowed heating of the reactor vessel (1) to 80 °C in order to achieve the evaporation of the water/sacrificial mixture. The vessel (1) was irradiated during 24 h (standard irradiation time) with a Hamamatzu Xe lamp with a spotlight placed at a distance of 5 cm above the reactor surface. The gas phase samples (500 µL) were collected with a Hamilton syringe and injected into the GC-TCD spectrometer. The quantification of the amount of H_2_ was determined after 24 h of irradiation. The molecular hydrogen peak area was calculated to the corresponding concentration using the standard calibration curve as reference. The micromoles of H_2_ produced were calculated taking into account the ideal gas law (n = PV/RT). Control experiments were done for the {Ta_6_Br^i^_12_}@GO-20L material (Appendix A) and confirmed the lowest amounts of gas production obtained. All the experiments showed the detection of H_2_ and the atmospheric gases, exclusively.

Reuse experiments were carried out for three cycles under the same conditions of the above experiments. The aqueous mixture at the end of the reaction was first removed, whereas the catalytic reactor with the cluster-based materials was purged under argon stream for 1 h. After, a fresh solution of H_2_O/MeOH/HBr was put on the reactor, and a second purge step was carried out in order to deoxygenate the system. This procedure was repeated for each reuse cycle. The percentage of the amount of H_2_ produced was calculated with respect to the value of hydrogen obtained in the first use.

## 3. Results and Discussion

In this research, the {Ta_6_Br^i^_12_}@GO hybrids were prepared from the [{Ta_6_Br^i^_12_}Br^a^_2_(H_2_O)^a^_4_]·4H_2_O and GO precursors. These nanostructured materials were characterized by textural, analytical, and spectroscopic techniques, and were applied as catalysts in the HER from water and light. The following two subsections encompass the results and discussion derived from this investigation.

### 3.1. Preparation and Characterization of Tantalum Materials

The surface of GO has different oxygen donor functionalities, mainly represented by hydroxyl, carboxyl, and epoxy groups, that are suitable for coordinative metal cluster anchoring [35,36]. Octahedral tantalum clusters are suitable candidates for immobilization since they have a high affinity towards oxygen-donor ligands. Brničević et al. reported that [{M_6_X^i^_12_}(H_2_O)^a^_4_X^a^_2_] (M = Nb, Ta; X = Cl, Br) compounds are capable to fully exchange the apical aqua ligands by aliphatic alcohols to provide [{M_6_X^i^_12_}(ROH)^a^_4_X^a^_2_] (R = CH_3_, C_2_H_5_, i-C_3_H_7_ and i-C_4_H_9_) products [40]. In this work, the immobilization of the [{Ta_6_Br^i^_12_}Br^a^_2_(H_2_O)^a^_4_] cluster onto GO functionalities to provide hybrid {Ta_6_Br^i^_12_}@GO nanomaterials was tackled.

The [{Ta_6_Br^i^_12_}Br^a^_2_(H_2_O)^a^_4_]·4H_2_O and GO precursors were first prepared following adapted reported procedures [24,37,38]. The high crystallinity and purity of [{Ta_6_Br^i^_12_}Br^a^_2_(H_2_O)^a^_4_]·4H_2_O were confirmed by SEM and Raman techniques (Appendix A). Preliminary immobilization experiments were run in ambient conditions by mixing [{Ta_6_Br^i^_12_}Br^a^_2_(H_2_O)^a^_4_]·4H_2_O with GO in hot THF, but cluster decomposition was observed. This was confirmed by SEM–EDS analyses of the solid, and through UV-vis analysis of the colored filtrate obtained after separation of the solid material. The EDS analysis showed a disproportion in the bromine (>0.4 wt %) and tantalum (*ca*. 0.8 wt %) content, which agrees with the decomposition of the cluster after reaction with GO in air. UV-vis analyses in milli-Q water solutions showed the disappearance of the band at 645 nm, a shift of the band at 750 nm to 715 nm, and a new band around 870 nm, which indicate full oxidation of the {Ta_6_Br^i^_12_}^2+^ to {Ta_6_Br^i^_12_}^3+^ species [26,27]. These results agree with the reported air-promoted cluster oxidation and decomposition towards Ta_2_O_5_ described for {M_6_X^i^_12_}^2+^ (M = Ta, Nb; X^i^ = halogen) cluster species [40,41].

The synthesis of the {Ta_6_Br^i^_12_}@GO nanohybrids was then performed under an inert atmosphere. Different tantalum cluster loadings were used in order to obtain the nanohybrids, namely: {Ta_6_Br^i^_12_}@GO-5L, {Ta_6_Br^i^_12_}@GO-20S, {Ta_6_Br^i^_12_}@GO-20L, and {Ta_6_Br^i^_12_}@GO-40L. The cluster immobilization yield was optimized by variying the reaction times from 2 to 16 h to afford the {Ta_6_Br^i^_12_}@GO-20S and {Ta_6_Br^i^_12_}@GO-20L materials, respectively. The cluster uptake for each solid was determined by ICP-AES analysis. The tantalum content in the materials obtained after longest reaction times (Ta (% *w*/*w*) was 2.58 for {Ta_6_Br^i^_12_}@GO-5L, 11.95 for {Ta_6_Br^i^_12_}@GO-20L, and 19.66 for {Ta_6_Br^i^_12_}@GO-40L). These values approximately match the expected [{Ta_6_Br^i^_12_}Br^a^_2_(H_2_O)^a^_4_]·4H_2_O cluster loadings, which indicate that the cluster immobilization approached quantitative results with longer reaction times. In contrast, a low tantalum content was found for the {Ta_6_Br^i^_12_}@GO-20S (1.54 % *w*/*w*) material. This result confirmed that the control of stirring time is important to allow the total of the cluster to be effectively anchored onto GO surfaces.

The PXRD patterns of the {Ta_6_Br^i^_12_}@GO nanomaterials were registered and compared with those of [{Ta_6_Br^i^_12_}Br^a^_2_(H_2_O)^a^_4_]·4H_2_O and GO precursors (Figure 2). The diffractogram of the {Ta_6_Br^i^_12_}@GO-5L hybrid shows the characteristic diffraction peaks of the GO, which are centered at 11.7° and 42.6° [42]. The hybrids with a higher amount of the tantalum cluster show a notable decrease in these peaks, attributed to the decrease in the already poor crystallinity of the GO layers. The remaining peaks correspond to crystalline [{Ta_6_Br^i^_12_}Br^a^_2_(H_2_O)^a^_4_]·4H_2_O, for which the diffraction patterns are indistinguishable in terms of the intensity peaks and 2θ values [26]. This indicates that the [{Ta_6_Br^i^_12_}Br^a^_2_(H_2_O)^a^_4_]·4H_2_O cluster material is supported onto GO sheets in its crystalline form.

The morphology of GO and chemical composition of the resulting {Ta_6_Br^i^_12_}@GO nanohybrids were examined by SEM and EDS techniques, respectively. The flaky texture of the GO (Figure 3a) reflects its characteristic layered microstructure and the larger interspaces of the layer and the thinner layer edges of GO can be clearly appreciated [43]. Regarding the {Ta_6_Br^i^_12_}@GO materials (Figure 3b–d), the {Ta_6_Br^i^_12_}^2+^ cluster crystals appear as small particles of lighter color with characteristic hexagonal shapes (Appendix A) that are homogeneously distributed onto the graphenic surface. It should be noted that, in the case of the materials with a higher content of the cluster, the crystals form aggregates as nanocrystals with a larger size.

In order to find more evidence for the presence of the {Ta_6_Br^i^_12_} cluster units in the {Ta_6_Br^i^_12_}@GO nanohybrids, several EDS spectra were recorded and analyzed in different regions of the sample (Appendix A). The results revealed that the material was composed uniquely of carbon, oxygen, bromine, and tantalum. Tantalum was exclusively detected in the brightest zones, and the averaged Ta/Br ratio found corresponds to 0.48, which matches the expected value (0.43) for the Ta_6_Br_14_ stoichiometry. The EDS analysis of the [{Ta_6_Br^i^_12_}Br^a^_2_(H_2_O)^a^_4_]·4H_2_O precursor shows both elements in the same proportion. These results prove the integrity of the GO-supported cluster units.

The type of immobilization of the cluster units onto GO surface was assessed by FTIR spectroscopy. The spectra of the nanohybrids, GO and [{Ta_6_Br^i^_12_}Br^a^_2_(H_2_O)^a^_4_]·4H_2_O precursors are depicted in Figure 4. The spectra of the graphenic materials show a wide band centered approximately at 3430 cm^−1^ that is associated with the stretching modes of the hydroxyl groups [44] that remain unaltered after cluster immobilization. The spectrum of a crystalline sample of [{Ta_6_Br^i^_12_}Br^a^_2_(H_2_O)^a^_4_]·4H_2_O shows water libration bands at 3040 cm^−1^ and 3243 cm^−1^, and appears in the fingerprint of the nanohybrids after cluster immobilization. Apart from the signal at 1627 cm^−1^ assigned to the bending vibration of water molecules, the spectrum of the pure tantalum cluster material does not show additional signals in the range of 500–2700 cm^−1^. Note that the characteristic {Ta_6_Br^i^_12_}^2+^ cluster core IR vibration bands at lower frequencies are not detectable within the registered spectral window [45].

The FTIR identification of {Ta_6_Br^i^_12_}@GO shows the two strong characteristic C=O vibration bands at 1726 and 1584 cm^−1^, and a shoulder at 1627 cm^−1^. The first band is shifted with respect to that of the GO and is associated with the carbonyl group vibration of the GO material, whereas the shoulder at 1627 cm^−1^, which corresponds to carboxylic/adsorbed water vibration bands, remains unaltered. A new and intense band appears at 1584 cm^−1^ after cluster immobilization. This band was also detectable in the spectrum of the (TBA)_2_Mo_6_I^i^_8_@GO nanohybrid and was attributed to the interaction of the hexametallic Mo_6_I^i^_8_ cluster units with the carboxylate functionalities of the graphene support. This band confirms, for the first time, the interaction of the {Ta_6_Br^i^_12_} cluster units with carboxylate funtionalities. In the {Ta_6_Br^i^_12_}@GO spectra, the hydroxyl vibration region characteristic of the GO appear distorted: the band at 1210 cm^−1^, which is characteristic of the C-OH graphene vibrations of GO, shifts to longer wavelenghts (1234–1244 cm^−1^) and increases in intensity. This peak is associated to a new Ta-O-C vibration of the coordinated carboxylate and alcoxo/alcohol groups. This is not surprising because octahedral tantalum clusters readily and rather selectively exchange apical ligands with alcohols [46,47] and alcoxo ligands [48,49], being the alcoxo cluster species commonly bonded to the oxidized {Ta_6_Br^i^_12_}^3+^ and {Ta_6_Br^i^_12_}^4+^ cluster cores. The epoxy region, however, is not altered: the intense band at *ca*. 1060 cm^−1^, which is assigned to the C-O-C vibration of GO, does not shift, and no new bands appear. It suggests that the tantalum cluster is not reactive enough to open the epoxy rings of the graphenic surface under experimental immobilization conditions. This behavior contrasts with that reported for the other cluster@GO ((TBA)_2_Mo_6_I^i^_8_@GO and (TBA)_2_Mo_6_Br^i^_8_F^a^_6_@GO) nanocomposites, which show changes in the epoxy/hydroxyl FTIR region of the GO [35,36]. The spectrum of the [{Ta_6_Br^i^_12_}Br^a^_2_(H_2_O)^a^_4_]·4H_2_O material features a band at 466 cm^−1^, that could be assigned to the Ta-O vibrations of Ta_2_O_5_ [50,51], which is present as an impurity in the [{Ta_6_Br^i^_12_}Br^a^_2_(H_2_O)^a^_4_]·4H_2_O precursor (1% *w*/*w* determined in aqueous solution by UV-vis spectroscopy). This band can be detected after cluster immobilization, but only in the spectra from the samples with higher tantalum loadings. The low intensity of this signal suggests that cluster degradation was efficiently prevented during and after the cluster immobilization reaction. These results unequivocally confirm that cluster immobilization occurred by partial ligand exchange between the aqua ligands and the carboxylic and hydroxyl groups of the GO surface to afford the coordinative anchoring of the {Ta_6_Br^i^_12_} cluster units.

The {Ta_6_Br^i^_12_}@GO hybrids and the crystalline precursor were also characterized by Raman spectroscopy. The most representative spectrum was measured under 785 nm for {Ta_6_Br^i^_12_}@GO-20L (Appendix A). The most intense bands are centered at 1342 cm^−1^ and 1598 cm^−1^ (G band), which correspond to the characteristic D and G graphenic bands, respectively [42]. Whereas the [{Ta_6_Br^i^_12_}Br^a^_2_(H_2_O)^a^_4_]·4H_2_O compound is highly active under 785 nm excitation at low Raman shifts, the nanohybrid show only two bands centered at 140 cm^−1^ and 153 cm^−1^ (Appendix A), which are assignable to vibrations of {Ta_6_Br^i^_12_} tantalum cluster units [45]. This behavior suggests that the GO strongly influences the electronic properties of the cluster thus indirectly indicating the coordinative nature of the cluster immobilization. This is also evident from Raman characterization of both materials under 514 nm: the nanohybrid shows a wide and low-intensity band at low Raman shifts (Appendix A), while the spectrum of the crystalline compound shows the characteristic pattern of the {Ta_6_Br^i^_12_} cluster compounds, with the most intense cluster bands at 122 cm^−1^, 172 cm^−1^, and 233 cm^−1^ [26], and with no additional bands at higher Raman frequencies. Similar Raman spectra were registered under N_2_ atmosphere, therefore, cluster decomposition under air was discarded.

The UV-vis-NIR DRS identification of the {Ta_6_Br^i^_12_}@GO materials also confirmed the presence of the metal cluster units. Figure 5 illustrates a comparison between the absorption spectra of the nanohybrids, [{Ta_6_Br^i^_12_}Br^a^_2_(H_2_O)^a^_4_]·4H_2_O, and GO materials. The spectra of {Ta_6_Br^i^_12_}@GO materials show a good correspondence with the most energetic absorption bands of the molecular material. In general, these bands are comparable to those assigned in the DRS and UV-vis spectra reported for {Ta_6_Br^i^_12_} cluster compounds and derived composites [16,17,18,19]. The cluster bands in the Vis–NIR region are most sensitive in the samples with the highest loadings of cluster material. The spectra of {Ta_6_Br^i^_12_}@GO-20S and 20 L materials show signals at 630 nm, 735 nm, and the most intense band at 890 nm. These bands are not detectable in the graphenic support and differ in shape and wavelength with respect to the molecular material. The bands detected at lower wavelengths (750 nm and 641 nm) for the crystalline [{Ta_6_Br^i^_12_}Br^a^_2_(H_2_O)^a^_4_]·4H_2_O cluster material correspond to the local excitations within the cluster core of the tantalum halogenated aqua clusters [25,26], and show a hypsochromic shift in the spectra of the nanohybrids due to the interaction of the {Ta_6_Br^i^_12_}^2+^ cluster core with the oxygen functionalities of the GO. A band at 850 nm in the spectrum of [{Ta_6_Br^i^_12_}Br^a^_2_(H_2_O)^a^_4_]·4H_2_O is associated with the effect of high crystallinity of the material under DRS acquisition conditions. The appearance of the intense band at 890 nm in the spectra of the nanohybrids, and the increase in the intensity of the band at 735 nm (relative to the above-mentioned cluster bands) with the reaction time, suggest the presence and accumulation of the oxidized {Ta_6_Br^i^_12_}^3+^ cluster species [16,17,18,19]. This is not unexpected, since previously reported 3+ and 4+ species coordinated to alcohol and alcoxo ligands have been isolated in ligand exchange reactions starting from the 2+ clusters in air [46,47,48,49]. In our case, the partial Ta_6_ cluster oxidation could be due to a redox reaction between the cluster and the oxygen functionalities on the GO surface during the immobilization process. Reaction time supports this hypothesis: the band at 890 nm appears at a lower relative intensity in {Ta_6_Br^i^_12_}@GO-20S with respect to {Ta_6_Br^i^_12_}@GO-20L spectra. In addition, it is known that the cluster oxidation is favored in solution in the presence of ketones under ambient conditions [16,17,18,19,26], and keto groups also form part of the graphenic support. Thus, in absence of O_2_, the presence of oxidized {Ta_6_Br^i^_12_}^3+^ cluster units is to be ascribed to the reduction in some of the graphenic functionalities by the cluster. Analyses of the materials from carbon suggest a partial reduction in the graphenic support after the immobilization process. The carbon content for the hybrid materials is *ca*. 12 % higher than that expected from pure GO and cluster mixtures in their corresponding ratios. This is in agreement with the ability of the {Ta_6_Br^i^_12_}^2+^ cluster species to be oxidized to the 3+ and 4+ species, and of GO to be reduced towards graphene *via* chemical or electrochemical means [52,53]. Alternatively, the reaction between [{Ta_6_Br^i^_12_}Br^a^_2_(H_2_O)^a^_4_]·4H_2_O and carboxylic and hydroxyl functionalities was assessed by UV-vis spectroscopy and in homogeneous conditions by using benzoic acid and phenol as respective models. UV-vis spectra show that after 24 h of reaction, no spectral changes of the cluster precursor material were detected (Appendix A). These results evidenced that the detection of {Ta_6_Br^i^_12_}^3+^ cluster species is exclusively ascribable to the redox coupling between O-functionalities supported onto GO sheets.

### 3.2. Photocatalytic Performance of Tantalum Nanomaterials

Gas phase photocatalytic water transformations constitute a promising alternative to more common liquid phase methodologies because soft conditions used in this case assure stability and recovery of the catalyst [35,36,54,55]. This also sets up a sustainable pathway to hydrogen production via electrolysis of moisture from air [56,57]. In our research, we investigated the catalytic performances of the {Ta_6_Br^i^_12_}@GO nanohybrids and the [{Ta_6_Br^i^_12_}Br^a^_2_(H_2_O)^a^_4_] and GO nanomaterials towards water reduction of H_2_ in the presence of vapor water mixtures and under photochemical conditions. Water was mixed with hydrobromic acid as the proton source and methanol as the sacrificial electron donor.

After 24 h of irradiation, all the materials selectively produced H_2_ but different activities were found (Figure 6a). Crystalline [{Ta_6_Br^i^_12_}Br^a^_2_(H_2_O)^a^_4_]·4H_2_O showed higher catalytic efficiency than unmodified GO. The performance of the crystalline cluster material, however, is low, due to its low surface area which prevents interaction between gas molecules and cluster sites. In contrast, the cluster immobilization onto GO sheets allows reactive species such as water, methanol, and H^+^ to be adsorbed on the more exposed surface of the material, and to promote the reaction, giving superior H_2_ yields. The {Ta_6_Br^i^_12_}@GO-20L nanomaterial provided the highest hydrogen production (84 µmol/g_cluster_) among all the nanohybrids and controls tested. This amount of H_2_ is superior to the sum of the yields achieved on the individual counterparts, which proves both the synergetic effect and the true hybrid nature of this nanomaterial. In fact, the activity of the cluster sites of {Ta_6_Br^i^_12_}@GO-20L and the molecular materials is 706 µmol·g^−1^ vs. 101 µmol·g^−1^ atomic tantalum (Figure 6b). This indicates that the performance of {Ta_6_Br^i^_12_}@GO-20L is seven-fold higher than for the microcrystalline [{Ta_6_Br^i^_12_}Br^a^_2_(H_2_O)^a^_4_]·4H_2_O, which demonstrates the enhancement of the catalytic efficiency of the cluster site supported onto the graphenic surface.

The activities found for the other nanohybrids are lower, with 32 µmol/g_cluster_, 55 µmol/g_cluster_, and 67 µmol/g_cluster_ for {Ta_6_Br^i^_12_}@GO-20S, {Ta_6_Br^i^_12_}@GO-5L, and {Ta_6_Br^i^_12_}@GO-40L, respectively. The {Ta_6_Br^i^_12_}@GO-20L provides the highest H_2_ production, while the {Ta_6_Br^i^_12_}@GO-5L and -20S materials, with the lowest tantalum loadings, are catalytically less active. Interestingly, the amount of supported cluster and the catalytic yields are not always proportional and the activity diminishes with the highest (*ca*. 40% *w*/*w*) tantalum loadings. This may be due to the too-large size of the crystallites of the tantalum cluster in {Ta_6_Br^i^_12_}@GO-40L, or to an agglomeration of the same, which limits the occurrence of successful charge transfer between the cluster and the GO, thus decreasing the effectiveness of the photocatalytic process.

It is worth stressing that these catalytic yields are superior to those reported for similar metal cluster-anchored GO nanocomposites, that is, for (TBA)_2_Mo_6_I^i^_8_@GO (30 µmol·g^−1^) [35] and (TBA)_2_Mo_6_Br^i^_8_@GO (<60 µmol·g^−1^) [36] in vapor or liquid phases, respectively, and with the same sacrificial reductant, whereas the reaction rate is in the same order of magnitude (Appendix A). The optimal values obtained with {Ta_6_Br^i^_12_}@GO nanocomposites also exceed the best value, achieved with [{Ta_6_Br^i^_12_}Br^a^_2_(H_2_O)^a^_4_]·4H_2_O (45 µmol·g^−1^) in aqueous HCl under non-catalytic conditions [27]. It is worth stressing that the highest catalytic activity found in this work (4 µmol·g^−1^·h^−1^) is of the same order of magnitude as that reported in the literature for other tantalum photocatalysts, such as MTaO_3_ (M = Li, Na, Mg) and MTa_2_O_6_ (M = Mg, Ba) tantalates [58], used in the heterogeneous system and in the liquid phase in water/methanol mixtures without co-catalysts (Appendix A). To the best of our knowledge, there is no previous work associated with the photocatalytic performance of tantalum–GO composites towards water reduction.

A plausible photosensitation mechanism can be proposed considering the low optical band gap (1.44 eV) of the {Ta_6_Br^i^_12_}^2+^ cluster associated to its good visible–NIR cluster absorbance, which is responsible for efficient utilization of solar energy. The energy positioning of the LUMO orbitals (−3.48 eV) of the cluster complex [59] guarantees the catalytic role of the cluster unit for water reduction reaction from the thermodynamic point of view. The role of GO as co-catalyst is based on its semiconductor nature, due to the presence of conductive sp^2^ and nonconductive sp^3^ carbon domains on the GO surface [60,61,62]. The band gap of GO (2.9–3.7 eV) depends on the coverage, arrangement, and relative ratio of the epoxy and hydroxyl groups [28]. Even if some partial reduction in GO occurs, the positioning of the conduction band (CB) of GO is practically independent of the degree of oxidation (from −0.52 and −0.75 eV vs Ag/AgCl) [61,63]. The ability to separate the photogenerated electron–hole pair, superior electron mobility, high surface area, and easy surface chemical modification make GO and GO-based materials excellent photocatalysts for water reduction under UV and visible irradiation [29,64]. Taking into consideration these characteristics, the coordinative anchoring of the Ta_6_Br^i^_12_ cluster units onto GO sheets is expected to improve the performance of both components in visible light region, as we have proved experimentally. Upon excitation by light, the HOMO–LUMO transition of the cluster compound reaches the excited state as the first step of cluster activation [27]. The {Ta_6_Br^i^_12_}^2+^/{Ta_6_Br^i^_12_}^3+^ system acts as an electron shuttle and, in the following step, injects two electrons into the CB of GO, which are invested in the reduction of water to provide H_2_ (Figure 7). Although we have not yet conducted mechanistic studies, the electron transfer from methanol into the valence band of GO would be feasible and provide necessary electrons to reduce the oxidized Ta_6_ cluster thus closing the catalytic cycle. The role of GO, therefore, is to enhance the efficiency of the electron transfer from cluster sites to water molecules and to prevent electron recombination. This behavior is associated with the coordinative binding of the cluster sites and the high exposition of cluster sites anchored onto GO sheets. The improvement of the catalytic performance in the nanohybrids, in comparison with the tantalum cluster and GO separately, is a consequence of the high charge mobility and surface area of GO, which are responsible of the synergetic behavior between the cluster and GO.

The recyclability of {Ta_6_Br^i^_12_}@GO-20L was tested in three consecutive runs under the same reaction conditions and the results are represented in Figure 8. The material proved to be stable after three reuse cycles, and the H_2_ production decreased slightly in the first recycling cycle (*ca*. 10%), but in the following two reuse cycles it seemed to remain stable with minimum variation, demonstrating the recyclability of the material.

## 4. Conclusions

Nanostructured {Ta_6_Br^i^_12_}@GO hybrids were synthesized in a single step from GO and [{Ta_6_Br^i^_12_}Br^a^_2_(H_2_O)^a^_4_]·4H_2_O precursors in THF under inert conditions. The immobilization was achieved with different cluster:GO ratios and reaction times, providing four different nanohybrids, with materials obtained with longer reaction times being more effective. The integrity of the cluster units was preserved after immobilization, as proven by ICP-AES and EDS. The PXRD and SEM identifications of the nanohybrids show high crystallinity of [{Ta_6_Br^i^_12_}Br^a^_2_(H_2_O)^a^_4_]·4H_2_O phases immobilized onto GO layers. Cluster material was anchored through carboxyl and hydroxyl functionalities as evidenced by FTIR and Raman analyses. UV-vis-NIR identification of the samples showed both the presence of {Ta_6_Br^i^_12_}^2+^ and {Ta_6_Br^i^_12_}^3+^ cluster units, with the latter ascribable to a partial oxidation of the material associated with a reduction in the functionalities of GO during cluster immobilization and to the coordinative coupling between inorganic–organic materials.

Selective photoreduction of water into H_2_ was achieved with {Ta_6_Br^i^_12_}@GO hybrids as catalysts. All the nanohybrids exhibited higher reaction yields than their precursors, which agrees with a synergetic effect between the cluster and graphenic counterparts. The enhancement of the photocatalytic behavior of the {Ta_6_Br^i^_12_} units and GO is due to the coordinative immobilization of the cluster onto the GO surfaces, which promotes the electron transfer from the photoexcited cluster to the GO. The best catalytic performance corresponds to the hybrid with 20% (*w*/*w*) cluster tantalum loading, and its efficiency is comparable to similar molybdenum cluster GO nanocomposites. The recyclability and the high output of this material were proven for three consecutive runs. The simplicity of the preparation of the nanohybrids and the photocatalytic design, the stability and recovery of the catalyst, and the efficient recycling make the developed methodology superior and more advantageous for converting water into H_2_.

## Figures and Tables

**Figure 1 nanomaterials-12-03647-f001:**
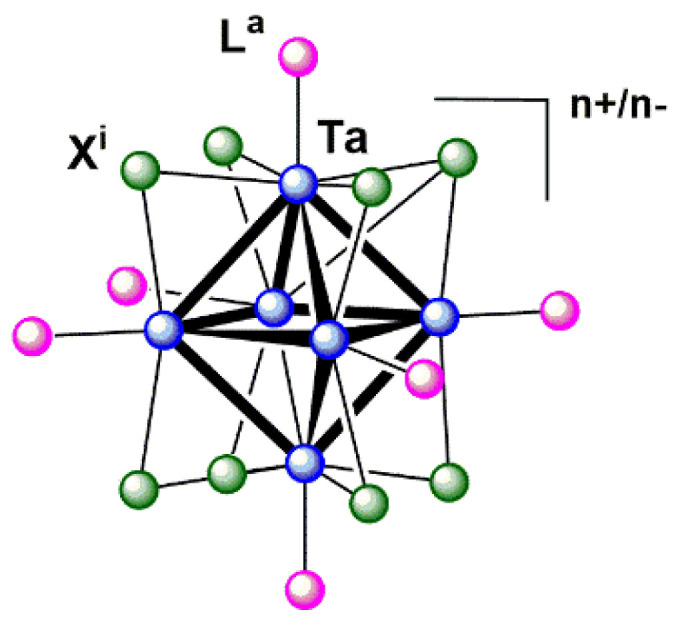
General representation of the octahedral [{Ta_6_X^i^_12_}L^a^_6_]^n^^+/−^ (circles in blue = Ta; green = X^i^; pink = L^a^) cluster unit.

**Figure 2 nanomaterials-12-03647-f002:**
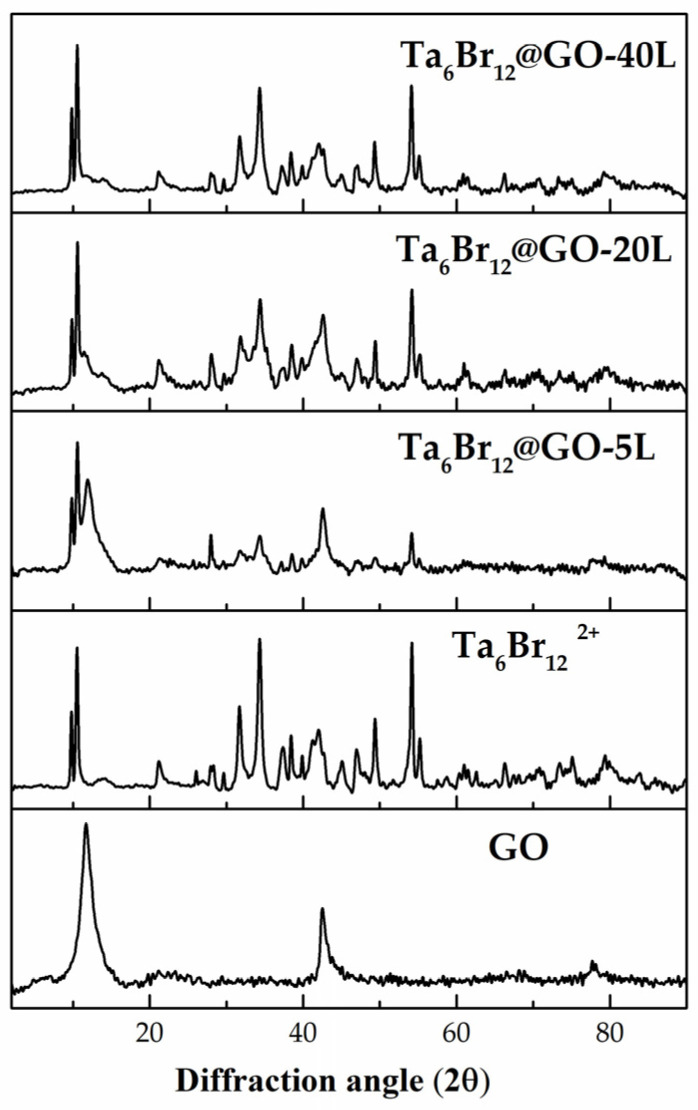
PXRD diffractograms of {Ta_6_Br^i^_12_}@GO, [{Ta_6_Br^i^_12_}Br^a^_2_(H_2_O)^a^_4_]·4H_2_O and GO materials.

**Figure 3 nanomaterials-12-03647-f003:**
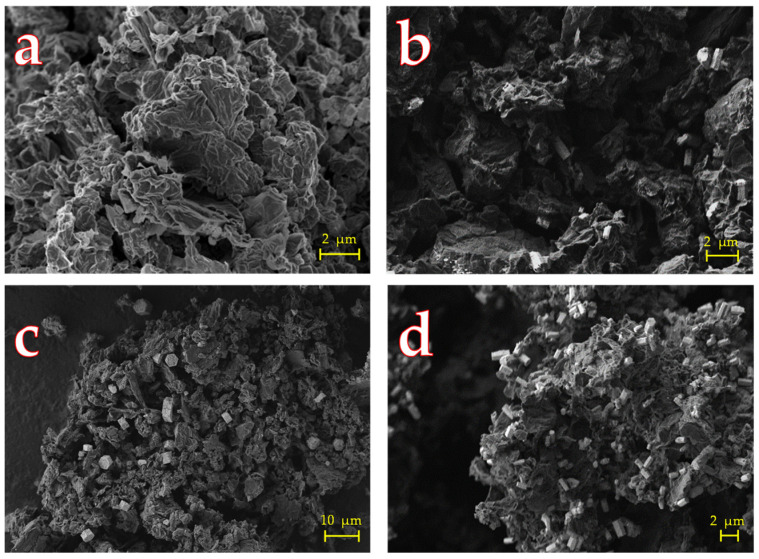
SEM micrograph of GO and {Ta_6_Br^i^_12_}@GO materials: (**a**) GO, (**b**) {Ta_6_Br^i^_12_}@GO-5L, (**c**) {Ta_6_Br^i^_12_}@GO-20L, and (**d**) {Ta_6_Br^i^_12_}@GO-40 L.

**Figure 4 nanomaterials-12-03647-f004:**
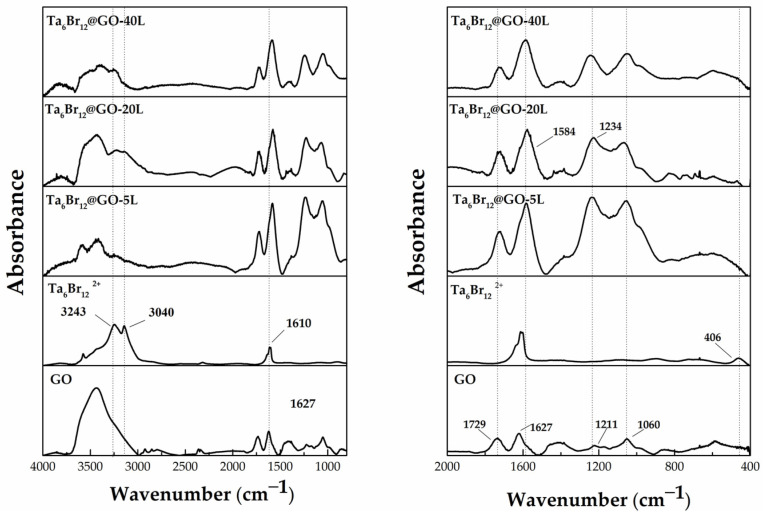
Two regions of the FTIR spectra of {Ta_6_Br^i^_12_}@GO, [{Ta_6_Br^i^_12_}Br^a^_2_(H_2_O)^a^_4_]·4H_2_O, and GO materials.

**Figure 5 nanomaterials-12-03647-f005:**
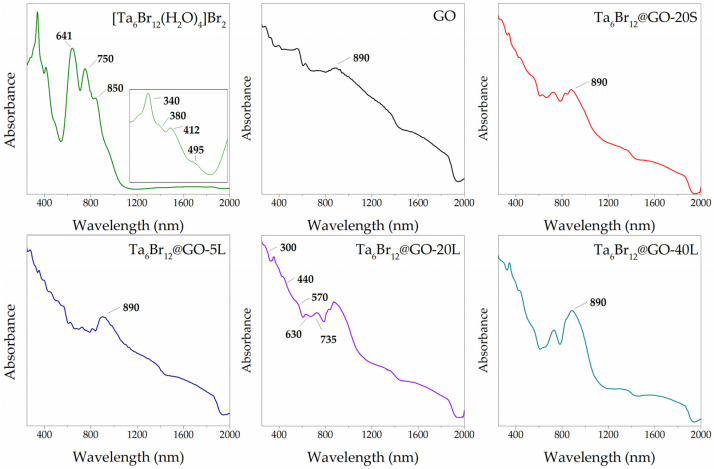
UV-vis DRS spectra of {Ta_6_Br^i^_12_}@GO, [{Ta_6_Br^i^_12_}Br^a^_2_(H_2_O)^a^_4_]·4H_2_O and GO materials.

**Figure 6 nanomaterials-12-03647-f006:**
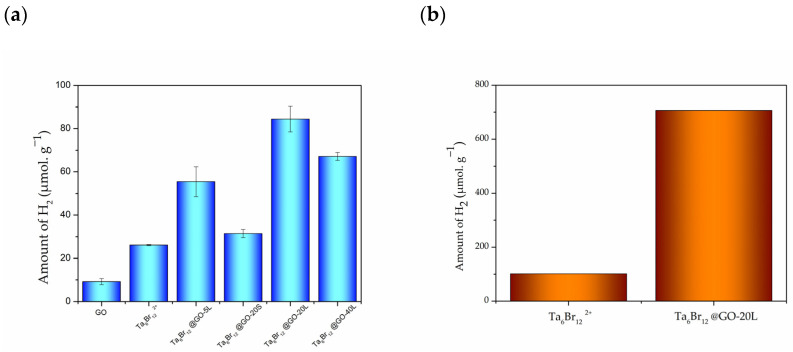
Amount of H_2_ evolved during the photocatalytic tests with respect to: (**a**) GO, microcrystalline molecular tantalum compound and nanohybrids catalysts; (**b**) tantalum content of the [{Ta_6_Br^i^_12_}Br^a^_2_(H_2_O)^a^_4_] and {Ta_6_Br^i^_12_}@GO-20L catalysts.

**Figure 7 nanomaterials-12-03647-f007:**
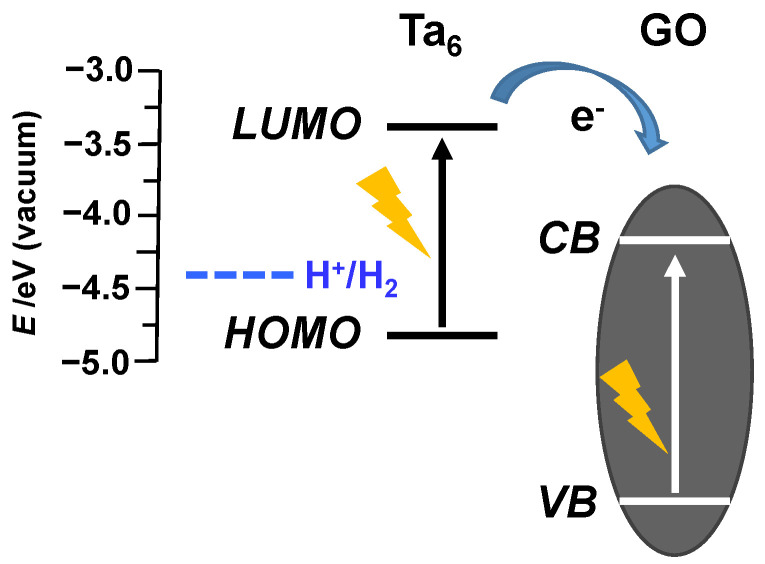
Schematic energy diagram which depicts the electron injection from the LUMO orbitals of [{Ta_6_Br^i^_12_}(H_2_O)^a^_6_]^2+^ cluster into the CB of GO.

**Figure 8 nanomaterials-12-03647-f008:**
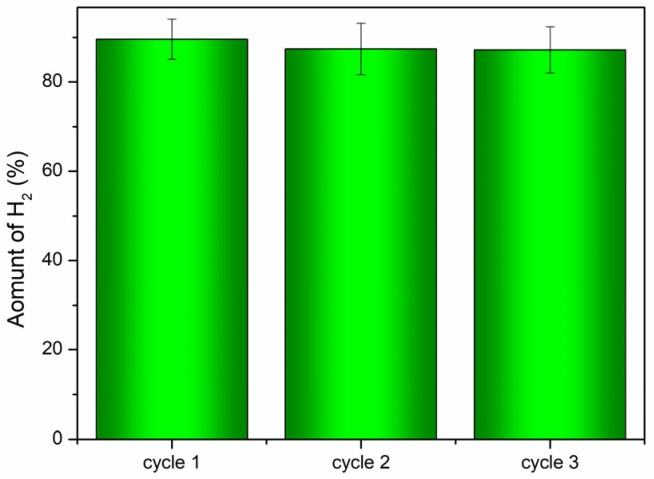
Recycling of the {Ta_6_Br^i^_12_}@GO-20L photocatalyst in the photochemical H_2_ production from water/methanol/HBr in vapor phase after 24 h of irradiation.

## Data Availability

Not applicable.

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
