# Peer review of "Nanostructured Hybrids Based on Tantalum Bromide Octahedral Clusters and Graphene Oxide for Photocatalytic Hydrogen Evolution"

_nanomaterials, 2022, doi:10.3390/nano12203647_

Round 1

Reviewer 1 Report

The authors developed a hybrid catalyst based on tantalum bromide octahedral clusters and graphene oxide. The hybrid catalyst showed higher catalytic efficiency than unmodified GO. The amount of H2 obtained is superior to the sum of the yields achieved on the individual counterparts, which proves both the synergetic effect and the hybrid nature of this nanomaterial. The hydrogen generation varies with the proportion of the cluster on the GO support. The results are interesting while the organization of data requires further improvement. Here are several questions and suggestions:

1. The abstract part lacks the research motivation. The authors are suggested to add 1-2 sentences about difficulties or problems in this field and explain why their work is valuable.

2. The {Ta6Bri12}@GO-20L is supposed to co more {Ta6Bri12}2+ cluster than that of {Ta6Bri12}@GO-5L. However, there is no big difference between figure 3c and figure 3b, please explain.

3. The authors applied black spectra in figure 4 left and colored spectra in figure 4 right. Please unify.

4. Some of the axes in figure 5 are missing. Please check.

5. It is strongly suggested to check the grammatical mistakes in the manuscript. For example, in line 298 “Is important to point out that…” and in line 515 “being longuest times…”

Author Response

See the attached file, please

Reviewer 2 Report

I have carefully read this paper entitled with “Nanostructured hybrids based on Tantalum Bromide Octahedral Clusters and Graphene Oxide for Photocatalytic Hydrogen Evolution". its interesting work, as a result, I have only a few minor points that the authors should address before it is accepted for publication. Please, publish subject to the following revisions:

1-Rewrite the novelty statement at the end of the introduction section.

2- Authors should justify the importance of the current work of how it is different from earlier reports. So, it’s better to add comparison table material and its performance to show the importance of the manuscript.

3-How do authors plot the energy diagram (Figure 7.), are they measure work functions to find HOMO and LUMO orbitals?

4-The reviewer suggested authors a more comprehensive background of Graphene oxide (GO) and the effect of photocatalytic degradation should be cited in the introduction for a wider readership. For example, some papers listed below:

- Photocorrosion Suppression and Photoelectrochemical (PEC) Enhancement of ZnO via Hybridization with Graphene Nanosheets. Applied surface science, Volume 502, 1 February 2020, 144189.

- Reduced Graphene Oxide (RGO) on TiO2 for an Improved Photoelectrochemical (PEC) and Photocatalytic activity. Solar Energy 190 (2019) 185–194,

- ZnO quantum dots- graphene composites: Formation mechanism and enhanced photocatalytic activity for degradation of methyl orange dy. Journal of Alloys and Compounds. Volume 663, 5 April 2016, Pages 738-749.

- Novel Visible Light Photocatalytic and Photoelectrochemical (PEC) Activity of Carbon-doped Zinc Oxide/Reduced Graphene Oxide: Supercritical Methanol Synthesis with Enhanced Photocorrosion Suppression. Journal of Alloys and Compound. Volume 723, 5 November 2017, Pages 1001-1010

Author Response

See the attached file, please
